# Bactericidal Activity of Ceragenin in Combination with Ceftazidime, Levofloxacin, Co-Trimoxazole, and Colistin against the Opportunistic Pathogen *Stenotrophomonas maltophilia*

**DOI:** 10.3390/pathogens11060621

**Published:** 2022-05-26

**Authors:** Paulina Paprocka, Angelika Mańkowska, Karol Skłodowski, Grzegorz Król, Tomasz Wollny, Agata Lesiak, Katarzyna Głuszek, Paul B. Savage, Bonita Durnaś, Robert Bucki

**Affiliations:** 1Department of Microbiology and Immunology, Institute of Medical Science, Collegium Medicum, Jan Kochanowski University, IX Wieków Kielc 19A, 25-317 Kielce, Poland; paulina.paprocka@ujk.edu.pl (P.P.); angelika.mankowska@ujk.edu.pl (A.M.); grzegorz.krol@ujk.edu.pl (G.K.); agatalesiak@wp.pl (A.L.); katarzyna.gluszek@ujk.edu.pl (K.G.); bonita.durnas@ujk.edu.pl (B.D.); 2Department of Medical Microbiology and Nanobiomedical Engineering, Medical University of Białystok, Jana Kilińśkiego 1 Białystok, 15-089 Białystok, Poland; karol.sklodowski@umb.edu.pl; 3Holy Cross Oncology Center of Kielce, Artwińskiego 3, 25-734 Kielce, Poland; tomasz.wollny@onkol.kielce.pl; 4Department of Chemistry and Biochemistry, Brigham Young University, Provo, UT 84602, USA; pbsavage@chem.byu.edu

**Keywords:** *Stenotrophomonas maltophilia*, ceragenins, synergy, trimethoprim/sulfamethoxazole

## Abstract

Background: *Stenotrophomonas maltophilia* (*S. maltophilia*) is an emerging opportunistic Gram-negative rod causing nosocomial infections predominantly in immunocompromised patients. Due to its broad intrinsic resistance to antibiotics, including carbapenems and the ability to form a biofilm, it is difficult to eradicate. Methods: In this study, the benefit of combined administration (potential synergism) and anti-biofilm activity of ceragenins: CSA-13, CSA-44, and CSA-131 (synthetic mimics of natural antimicrobial peptides) with ceftazidime, levofloxacin, co-trimoxazole and colistin against clinical strains of *S. maltophilia* were determined using MIC/MBC (minimum inhibitory concentration/minimum bactericidal concentration), killing assays and CV staining. Results: Obtained data indicate that the ceragenins exhibit strong activity against the tested strains of *S. maltophilia* grown in planktonic culture and as stationary biofilms. Moreover, with some strains, the synergy of ceragenins with conventional antibiotics was observed Conclusion: Our data suggest that ceragenins are promising agents for future development of new methods for treatment of infections caused by *S. maltophilia*, along with its potential use in combination with conventional antibiotics.

## 1. Introduction 

*S. maltophilia* is a motile, glucose non-fermentative, catalase-positive, usually oxidase-negative, multidrug-resistant Gram-negative rod. Since its first isolation in the 1940s, its systematic name has changed several times (from *Bacterium bookeri*, to *Pseudomonas maltophilia*, later to *Xanthomonas maltophilia* until the current name *Stenotrophomonas maltophilia)* [1]. *S. maltophilia* is widely distributed in natural environments. It can be isolated from water sources, soil, plants and animal microbiota. It was also described as able to exist in some free-living amoeba genera [2]. In hospital settings, *S. maltophilia* can be isolated from various, especially humid niches (tap water, showerheads, faucets, contaminated handwashing soaps, disinfectants, hemodialysis water), and medical devices such as nebulizers, renal units, and catheters [3]. Improperly cleaned and sterilized fiberoptic bronchoscopes were also described as potential sources of these bacteria. *S. maltophilia* present on staff’s hands, in the cleaning tank and on the inner wall of the bronchoscope pose a risk of nosocomial infections. Thus, strict adherence to cleaning, sterilization and hand hygiene procedures is essential to prevent and control the spread of *S. maltophilia* in hospital settings [4].

*S. maltophilia* was considered a non-pathogenic microorganism for many years, but during the last two decades, its role as an opportunistic pathogen is constantly increasing. In general, this microorganism is among the four most common, after *Pseudomonas aeruginosa, Acinetobacter baumannii* and *Burkholderia cepacia* complex, as a non-fermenting bacillus causing infections in humans [5]. *S. maltophilia*, as an opportunistic pathogen, is not equipped with many virulence factors. It can produce enzymes (elastase, hyaluronidase, lipase, protease, RNase and DNase), lipopolysaccharide (LPS) [6] and factors related to the production of biofilm such as fimbriae type 1 [7] or DSF (Diffusible Signal Factor) [8]. *S. maltophilia*, similar to many other microbes, has the ability to form biofilm both on artificial surfaces such as plastic or glass materials, as well as on biotic ones. This bacterium can also produce biofilm together with other bacterial species. It also produces the DSF factor, which is responsible for the production of biofilm [9]. Immunocompromised individuals, including patients in the Intensive Care Unit (ICU), people with cancer, patients with organ failure, patients with cystic fibrosis (CF) [10,11], patients with COPD (Chronic Obstructive Pulmonary Disease) [12], bronchiectasis, lung transplant recipients [13] and neonates [14] are all vulnerable to infections caused by this microorganism. Other major risk factors of a severe infection include long hospitalization, previous carbapenem therapy, neutropenia and lung dysfunction [15]. The most common clinical forms of infections are blood infections in neutropenic patients, pneumonia, [7] especially respiratory-associated pneumonia, and urinary tract infections in catheterized patients [16]. Occasionally, *S. maltophilia* only causes colonization and may be present in respiratory aerosols or mucus, urine and wound exudates of these hosts [17]. 

The most important factor contributing to the spread of *S. maltophilia* in the hospital environment is its high intrinsic resistance to antibiotics such as: penicillins, cephalosporins, carbapenems, aminoglycosides, and macrolides [18]. The mechanisms of resistance are as follows: low membrane permeability, antibiotic-degrading or modifying enzymes, and efflux pumps removing antibiotics from the bacterial cell [19,20]. From the clinical point of view, the most important resistance mechanism of *S. maltophilia* to β-lactams is the production of two β-lactamases: L1 and L2. The L1 enzyme is a metalo- β-lactamase with activity against almost all β-lactam antibiotics (penicillins, cephalosporins and carbapenems), but not against monobactams. The L2 enzyme shows cephalosporinase activity but can hydrolyze aztreonam and is susceptible to β-lactamase inhibitors. *S. maltophilia* also exhibits intrinsic resistance to aminoglycosides. It is associated with various mechanisms (including mutations within the 16S rRNA subunit, active pumping of antibiotics out of the cell (eflux) and changes in the structure of antibiotics through various enzymes) [21].

Due to intrinsic resistance, the choice of antibiotics to treat *S. maltophilia* infections is limited to co-trimoxazole, minocycline, levofloxacin, ciprofloxacin, ticarcillin with clavulanic acid, and ceftazidime; Ref. [18] in some cases, piperacillin with tazobactam, doxycycline, and colistin were used [22]. Among newer antibiotics, tygecycline can be a potential therapeutic option [23]. The drug of choice is trimethoprim/sulfamethoxazole (co-trimoxazole; SXT) used in high doses [24]. 

Acquired resistance develops through horizontal gene transfer or mutations and includes, for example, the acquisition of class 1 integrons and ISCR elements that are responsible for resistance to SXT, as well as multi-drug pumps and antibiotic-modifying enzymes [19,20]. Efflux pumps and the presence of the *qnr* gene, which make gyrase and topoisomerase IV non-sensitive to quinolones, contribute to the development of quinolone resistance. The *sul1*, *sul2* and *dfrA* genes are responsible for resistance to SXT. As for the efflux pumps in *S. maltophilia*, five families were identified. These are, among others, the resistance-nodulation-cell-division (RND) family, major facilitator superfamily (MFS), small multidrug-resistance (SMR) family, ATP-binding cassette (ABC) family and the multidrug and toxic compound extrusion (MATE) family. All these mechanisms can protect *S. maltophilia* against the action of antibiotics [25]. Acquired resistance extends the *S. maltophilia* resistance profile to other antibiotics and makes the treatment of infections caused by these bacteria a challenge. Although recently several new antibiotics were approved for use in hospital practice, only cefiderocol and eravacycline display activity against *S. maltophilia*. Cefiderocol is a new drug used to combat SXT-resistant *S. maltophilia* strains and is also effective in treating infections caused by bacteria-producing carbapenemases [26]. Eravacycline is a synthetic tetracycline that is active against Gram-negative pathogens, including carbapenem-resistant *S. maltophilia* [27]. Therefore, the search for new antibiotics with activity against Gram-negative rods including *S. maltophilia* is an urgent need. 

In this context, in recent years, ceragenins (CSAs) have been investigated as broad-spectrum antibacterial agents [28]. These compounds were designed to mimic antimicrobial peptides (CAPs) but, due to their structure, they are resistant to the action of proteases typically present at the sites of infection. They show antibacterial and antiviral activity as well as activity against parasites. The interaction between positively charged ceragenins and negatively charged microbial membranes leads to membrane dysfunction (depolarization and increase in permeability) which translates to bacterial death [29,30]. Ceragenins also display anti-biofilm activity [31] and bacteria do not acquire resistance to these compounds even after extended passages [32]. However, such a possibility should always be considered. Additionally, the use of ceragenins in higher doses might cause a toxic effect and hemolysis towards host cells. To limit this effect, they can be delivered to targeted microorganisms as magnetic nanoparticles (MNP) cargo or in combination with pluronic [33]. 

Here, we assessed the susceptibility of selected clinical strains of *S. maltophilia* to ceragenins: CSA-13, CSA-44 and CSA-131, conventional antibiotics SXT, ceftazidime (CAZ), levofloxacin (LVX) and colistin (COL) and their combinations. In the case of some *S. maltophilia* strains, we observed a synergistic effect of ceragenins in combination with some conventional antibiotics. These results suggest the possibility of new treatment options against *S. maltophilia* that might be developed with the implementation of ceragenins alone or in combination with conventional antibiotics. 

## 2. Results

### 2.1. Susceptibility of S. maltophilia Strains to Ceragenins and Conventional Antibiotics

The values of MIC (minimal inhibitory concentration) and MBC (minimum bactericidal concentration) of the tested compounds against selected *S. maltophilia* strains are presented in Table 1. These data show that out of the six clinical strains, five showed resistance (R) to ceftazidime, and one was resistant to levofloxacin. With co-trimoxazole, which is the drug of choice, all strains were classified as susceptible to increased exposure (SIE). Among all tested strains, three were resistant to colistin. The MIC values were interpreted according to EUCAST (European Committee on Antimicrobial Susceptibility Testing) recommendations [34]. The reference strain displayed resistance to ceftazidime and colistin but it was sensitive to levofloxacin and co-trimoxazole (SIE). It is worth noting that the MIC values for the tested ceragenins were in the same range for all the tested strains, regardless of their pattern of resistance to conventional antibiotics. More precisely, MIC values ranged between 0.5–2 µg/mL for CSA-13, 0.5–4 µg/mL for CSA-44 and 0.5–2 µg/mL for CSA-131. For all tested strains, MBC values determined for ceragenins followed the trend observed for their MIC values. Subsequent experiments were performed using five strains (numbers 1–5) for which a decrease in MIC values was observed after the use of CSA-131 in combination with SXT.

As shown in Table 2, the viability of strains resistant to ceftazidime and levofloxacin was strongly decreased when a combination of tested ceragenins (CSA-13, CSA-44, CSA-131) with currently used antibiotics was used as an intervention. Such combinations show a greater potential for the eradication of *S. maltophilia* than the use of individual antibiotics. This is most evident in the case of strains whose MIC values for CAZ and LVX are very high, but in combination with ceragenin, this value decreases and we classify them as susceptible and sensitive strains with increased exposure (ceftazidime 4(8)≤, levofloxacin 0.5(2)≤, EUCAST (CLSI)) (Table 2) [34]. It is worth noting that despite the fact that the synergistic effects resulting from the combination of tested ceragenins with selected antibiotics were not observed in all cases, no antagonistic effect was detected with all tested combinations (Table 3).

Ceragenin CSA-131 displayed the strongest activity against all tested strains of *S. maltophilia.* Among the ceragenins tested here, the lowest average MIC value determined for this collection of bacterial strains was obtained for CSA-131 (1.1 µg/mL) compared to CSA-13 (1.3 µg/mL) or CSA-44 (1.9 µg/mL); therefore, ceragenin CSA-131 was used for further studies (killing assay; activity against biofilm) to assess ceragenin activity in combination with SXT. The study shows that when CSA-131 is used individually, the bactericidal effect against tested strains of *S. maltophilia* was observed at the range of 0.7–2 µg/mL, while in the case of the combination of CSA-131 and SXT (1:1), it was at the range of 0.5–2 µg/mL (Figure 1). When CSA-131 was tested at a concentration of 0.5 µg/mL and SXT at varying concentrations (0.5–50 µg/mL) against strain 1, changes in bacterial outgrowth were not observed (Appendix A). 

### 2.2. Ability of CSA-131, SXT and CSA-131 + SXT to Prevent Biofilm Formation and to Disrupt Establish Stationary Biofilm

Our research shows that ceragenin CSA-131 alone, as well as in combination with SXT, inhibits the formation of biofilms with all tested *S. maltophilia* strains and at all tested time points (24, 48 and 72 h). These antimicrobial agents also display strong activity against established biofilm Figure 2A–C and Appendix A. CSA-131 was also able to target bacterial cells enveloped in exopolysaccharides of the biofilm matrix. Indeed, when tested against pre-formed biofilm, outgrow of *S. maltophilia* cells was decreased, especially in the case of the mature biofilm (72 h). CSA-131 effects against bacterial cells residing within the structure of mature biofilm were higher in combination with SXT, Figure 2D–F. Interestingly, our study also suggests that a “preventive effect” of CSA-131 against *S. maltophilia* biofilm formation is more pronounced when tested against the resistant strains (Appendix A).

### 2.3. Hemolytic Activity of CSA-131 in Combination with SXT 

Our research shows that the tested ceragenin, both alone and in combination with SXT, can cause hemolysis of parts of the red blood cells. However, this effect is observed only at doses that were more than 10 times higher than the MIC values. At different time points (1, 6 and 12 h), this effect was observed only at the dose of CSA-131 above 20 µg/mL. SXT, on the other hand, does not show hemolytic effects, even at the high tested doses (50 µg/mL), Figure 3.

## 3. Discussion

*S. maltophilia* infections can be very serious and sometimes even fatal. Currently, trimethoprim/sulfamethoxazole is the first recommended treatment option in case of those infections. Unfortunately, resistance to this antibiotic combination has emerged and continues to increase. Although all our strains tested here were susceptible to SXT, *S. maltophilia* resistance to SXT has already been reported in Poland (Małopolska and Silesia regions) [35] and is associated with the presence of genes *sul1*, *sul2* and *dfrA* [36,37]. In the last year, it was also found that fluoroquinolones (including levofloxacin) can exhibit efficacy similar to trimethoprim/sulfamethoxazole [38]. Good bactericidal effects were also obtained after the use of other antibiotics in combinations such as: SXT/CIP (ciprofloxacin), CAZ/LVX, TIM (ticarcillin-clavulanate)/SXT, TIM/ATM (aztreonam), TGC (tigecycline)/COL, COL/RD (rifampicin), CAZ/MH (minocycline), LVX/E (erythromycin), TGC (tigecycline)/FOS (fosfomycin) [39]. As evidenced by studies conducted by Biagi et al., an effective method for the treatment of infections caused by MDR (multidrug-resistant) *S. maltophilia* strains resistant to SXT and/or levofloxacin is the use of aztreonam-avibactam [40]. 

A combination therapy regimen to increase antimicrobial effects and delay the development of resistance may be ideal. The choice of monotherapy or combination therapy for the treatment of *S. maltophilia* is still a matter of controversy, but the use of several antimicrobial compounds shows better effects, especially in the most severely ill as well as in infections with resistant strains [41]. 

Limitations in treatment options in *S. maltophilia* infections necessitate the search for new antimicrobials. Ceragenins exhibit a wide range of antimicrobial activity [42] and may represent a promising future opportunity. The activity of these compounds against Gram-negative, glucose non-fermenting rods has already been assessed in previous studies. As noted by Nainu et al., CSA-131 is a good bactericidal and anti-biofilm agent against *Pseudomonas aeruginosa*, and in combination with colistin, it also acts against *Acinetobacter baumannii* CRAB (carbapenem-resistant) strains [43]. The latest research by Oyardi et al., carried out with 40 strains of *S. maltophilia* isolated from patients suffering from cystic fibrosis also indicates that CSA-131 and CSA-131P (delivered with Pluronik F-127) can be considered effective antibacterial and anti-biofilm agents against *S. maltophilia* [20]; results that agree well with our findings. In another study, Ozbek-Celik investigated the effects of CSA-13 and CSA-131 on *Klebsiella pneumoniae* strains resistant to meropenem and colistin. They found that ceragenins exhibited strong bactericidal activity [44], confirming that this class of molecules can be developed against Gram-negative rods which currently represent the greatest clinical challenge in the rational administration of antibiotics, especially in a hospital setting. Among the ceragenins tested here, CSA-131 showed the lowest average MIC (1.1 µg/mL) value among all tested *S. maltophilia* strains. This is most likely due to the long acyl chain of this ceragenin, permitting better penetration into bacterial cell external plasma membranes. Such conclusions were also drawn by Oyardi et al., demonstrating that the ceragenin CSA-131 can be used against *S. maltophilia*, both as an anti-bacterial and an anti-biofilm compound. Additionally, the authors showed that CSA-131 has low cytotoxicity to the IB3 -1 bronchial epithelial cell. Together these findings strongly indicate a need for future development of CSA-131 as a new therapeutic drug for patients with cystic fibrosis suffering from lung infections caused by Gram-negative rods including *S. maltophilia* [20].

Looking at the molecular background of ceragenin action, Chin et al. noted that the mechanisms of ceragenins and β-lactam antibacterial action are somewhat similar, considering their targets. In the case of ceragenins, they rely on the depolarization of the cell membrane, while β-lactam inhibits the synthesis of the bacterial cell wall. The detergent-like activity of ceragenins might additionally facilitate the diffusion of other antibiotics into bacterial cells, thus providing a pathway for the synergistic effects that have often been observed when ceragenins are combined with other antibiotics [45]. In our experimental setting, we observed four synergy effects and 11 partial synergy effects, most of which occurred with the use of ceragenin with the β-lactam antibiotic ceftazidime. In the case of other compounds, synergistic effects were less pronounced. Previously, synergistic effects were also reported for the combined administration of CSA-13 with CSA-131, CSA-13 with CSA-138 and CSA-131 with CSA-138 against carbapenem-resistant *Acinetobacter baumannii* (CRAB) strains [46], and for CSA-13 and lysozyme against *Bacillus subtilis* [47]. 

Considering that more than 60% of infections are associated with the ability of a microorganism to form a biofilm, much attention is also paid to research on the anti-biofilm properties of ceragenins. Biofilm is also an important factor in *S. maltophilia* infections, [48] and the anti-biofilm effects of ceragenins suggest that they have the potential for application against *S. maltophilia* strains [19,49]. The high anti-biofilm activity of ceragenins is confirmed by many previous studies [50,51,52]. For example, Chmielewska et al. demonstrated the effective action of CSA-13, CSA-44 and CSA-131 against biofilm produced by the NDM-1 (New Delhi metallo-β-lactamase-1) producing strains *Escherichia coli* BAA-2471, *Enterobacter cloacae* BAA-2468, and *Klebsiella pneumoniae* BAA-2472 I BAA-2473 [53]. 

A very important aspect in the study of new compounds is the evaluation of their biocompatibility. Our experiment confirmed previous studies on the safety of ceragenins when they are used in low doses [54]. The use of high doses of ceragenins, due to their effect on the membranes of red blood cells, may cause toxic effects; however, as studies show, these doses are much higher and exceed the MIC value needed to kill the bacteria [53].

Overall, our study results add to the collection of attributes of ceragenins that encourage the development of this class of compounds as new therapeutic solutions. The value of the proposed research is significant in the context of fighting infections caused by *S. maltophilia* as individual molecules and by reducing the MIC values of traditionally used antibiotics. 

## 4. Materials and Methods

### 4.1. Spectrum of Tested Bacteria

Antimicrobial tests were performed using 6 clinical and 1 standard ATCC 13636 (Manassas, VA, USA) *S. maltophilia* strains (indicated as strain 1 on the listed bacteria). Clinical strains were the isolate derived from patients of the Świętokrzyskie Cancer Center in Poland (strain 2 from-pharyngeal swab, strain 3 from-wound swab, strain 4 from-tongue swab, strain 5 from-voice prosthesis, strain 6 from-wound swab, strain 7 from-rectal swab). *Stenotrophomonas*-like colonies were isolated using Mac Conkey agar (Thermofisher Scientific, Waltham, MA, USA) and identified using the VITEK 2 Compact (bio-Mérieux, Marcy-l’Étoile, France). All *S. maltophilia* strains were banked in the MAST CRYOBANK system (Mast Diagnostics) and stored at −80 °C for further research. The following information was collected for each clinical isolate: the source of the material from which the microorganism was isolated, the patient’s age, sex, clinical diagnosis, and information on the resistance pattern of the isolated strain. The study was performed after approval by the Bioethics Committee of the Jan Kochanowski University in Kielce (No. 42/2021), and in accordance with the guidelines contained in the Helsinki Declaration. According to the decision of the Bioethics Committee the informed consent to perform these experiments was not necessary because the bacterial strains came from the laboratory collection. 

### 4.2. Determination of MIC and MBC

MIC measurements were performed using the EUCAST microdilution method [55]. For this purpose, the serial dilutions from 256 µg/mL to 0.125 µg/mL of the following antibacterial agent: trimethoprim/sulfamethoxazole (SXT ratio 1:19), ceftazidime, levofloxacin (Merck Life Science, Darmstadt, Germany) and colistin (Merlin, Bornheim, Germany) were prepared in a 96-well plate. Antibiotics from Sigma (Darmstadt, Germany) were prepared according to the manufacturer’s recommendations and dissolved in DMSO (Dimethylsulfoxide). For serial dilutions of compounds Mueller–Hinton Broth (MH broth, Thermofisher Scientific, Waltham, MA, USA) as used. Then 100 µL of the microorganism suspension with a final concentration of 10^5^ CFU/mL (colony-forming unit) was added to the medium. The MIC values were assessed visually. In order to determine the minimum bactericidal concentration, 10 µL of the suspension was transferred from the tested sample on the surface of the Luria Bertani agar (LB agar, BD Difco, Sparks, MD, USA). After 24 h of incubation at 37 °C, the presence of microbial colonies was assessed [56]. The same procedure was performed to determine the MIC and MBC values for CSA-13, CSA-44 and CSA-131 ceragenins [46]. 

### 4.3. Assessment of the Fractional Index of Inhibitory Concentration FICI/FIC 

In order to evaluate the synergistic effect of ceragenins (CSA 13, CSA-44 and CSA-131) in combination with conventional antibiotics (tested in a 1:1 ratio), the MIC was determined for these compounds for all tested strains using the microdilution method. 

To assess FICI/FIC (fractional inhibitory concentrations index) the following formula was used: FICa = MIC(ab)/MIC(a), FICb = MIC (ab)/MIC(b) and FICI = FICa + FICb (where: MICa, MICb are MIC values of compounds used separately, MICab-MIC values of compounds used in combination) [57,58]. The results were interpreted according to the obtained FICI data (<0.5 synergy (S), <0.76–0.5 is partial synergy (PS), 0.76 <FICI ≤ 1 addition (AD), 1.01 < FICI ≤ 4 is indifference (I), while FICI > 4 is antagonism (AN)) [59].

### 4.4. Killing Assay 

To determine the bactericidal activity of CSA-131, SXT and CSA-131 in combination with SXT (1:1), a killing assay was performed. A bacterial suspension of 10^5^ CFU/mL in PBS was added to the solution with tested compounds ranging from 0.01–2 µg/mL placed in a 96-well plate. After one hour of incubation at 37 °C, the plates were transferred to ice and the samples were diluted 10–1000 times. The test was performed in triplicate. An equal volume 10 µL from each condition was plated on LB (BD Difco, Sparks, MD, USA) and incubated overnight at 37 °C. Colony-forming units were determined from the dilution factor [30].

### 4.5. Prevention of Biofilm Formation and Disruption of Established Biofilms

The fluorometric method using resazurin (Sigma-Aldrich, Darmstadt, Germany) was used to assess the ability of tested compounds to prevent biofilm formation. After incubation of the tested bacteria in black bottom plates in the presence of CSA-131, SXT (1:19) and CSA-131/SXT (1:1) at concentrations (1–50 µg/mL) at 37 °C, for 24, 48 and 72 h, the plates were washed 3× with PBS to eliminate plankton cells and again incubated for one hour with resazurin at a concentration of 0.2 mg/mL. After this time, the fluorescence intensity at λ = 520–590 nm was measured using a Tecan Spark plate reader (Spark Control Magellan V2.2, Männedorf, Switzerland) [60]. 

After 24, 48 and 72 h of growth in LB (BD Difco, Sparks, MD, USA) at 37 °C, the establish stationary biofilm, was washed to eliminate the planktonic bacteria. Then CSA-131, SXT (1:19) and CSA-131/SXT (1:1) were added at concentrations (1–100 µg/mL) and incubated for 1h. The plates were then washed with PBS and placed in an ultrasonic bath (FVAT) for 15 minutes. After this time, the removed biofilm was diluted 10–1000 times and 10 µl from each well was plated on Mueller–Hinton Agar (MH agar, Thermofisher Scientific, Waltham, MA, USA). After 24 h incubation at 37 °C, the number of grown colonies was counted [53].

### 4.6. Hemolytic Activity 

A red blood cells (RBCs) hemolysis test was used to determine the toxicity. The study was performed after approval by the Bioethics Committee of the Jan Kochanowski University in Kielce (No. 42/2021), and in accordance with the guidelines contained in the Helsinki Declaration. Human RBCs were isolated from the blood of healthy volunteers. All donors provided written consent. The activity of the tested compounds: CSA-131, SXT (1:19) and CSA-131/SXT (1:1) was assessed at a concentration range 1–50 µL and incubated with red blood cells (hematocrit~5%) for 1, 6 and 12 h in 37 °C. Then, the samples were centrifuged (2500× *g* for 10 min.) and the amount of released hemoglobin was measured colorimetrically at λ = 540 nm using a Tecan Spark plate reader (Spark Control Magellan V2.2, Austria). A 1% Triton X-100 (Sigma Aldrich, Darmstadt, Germany) was used as a positive control (100% hemolysis) [61].

### 4.7. Statistical Analysis 

Graph Pad Prism version 8 (San Diego, CA, USA) was used for statistical analyses. Data collected are presented as the mean from 3–6 experiments. The Student’s test was used for the analysis of significance, and the value of ≤0.05 was considered statistically significant.

## 5. Conclusions

The growing resistance of bacteria to currently used antibiotics requires increasing effort to look for alternative substances and new strategies to combat them. Our data show that ceragenins CSA-13, CSA-44 and CSA-131 have great potential as bactericidal agents against *S. maltophilia,* including strains resistant to currently used antibiotics. Future studies, including those using animal models, are needed to confirm our in vivo observations.

## Figures and Tables

**Figure 1 pathogens-11-00621-f001:**
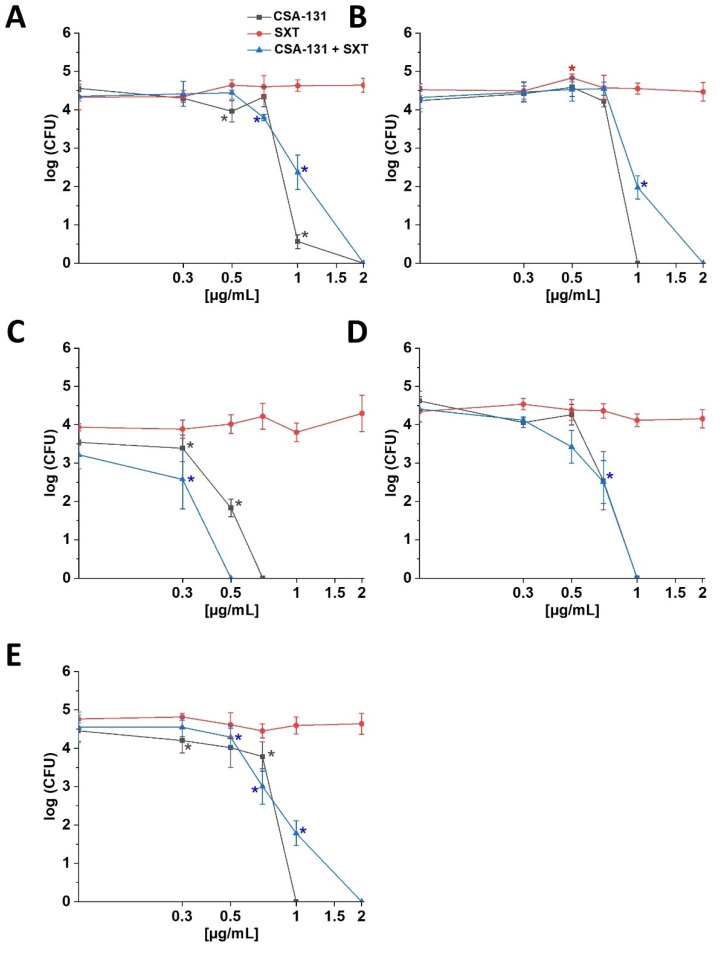
Bactericidal activity of CSA-131 (black squares), SXT (red circles), and CSA-131 + SXT (blue triangles) against *S. maltophilia* (strains 1, 2, 3, 4 and 5; panel (**A**), (**B**), (**C**), (**D**) and (**E**), respectively). Results show mean ± SD from six measurements. * indicates statistical significance at ≤0.05.

**Figure 2 pathogens-11-00621-f002:**
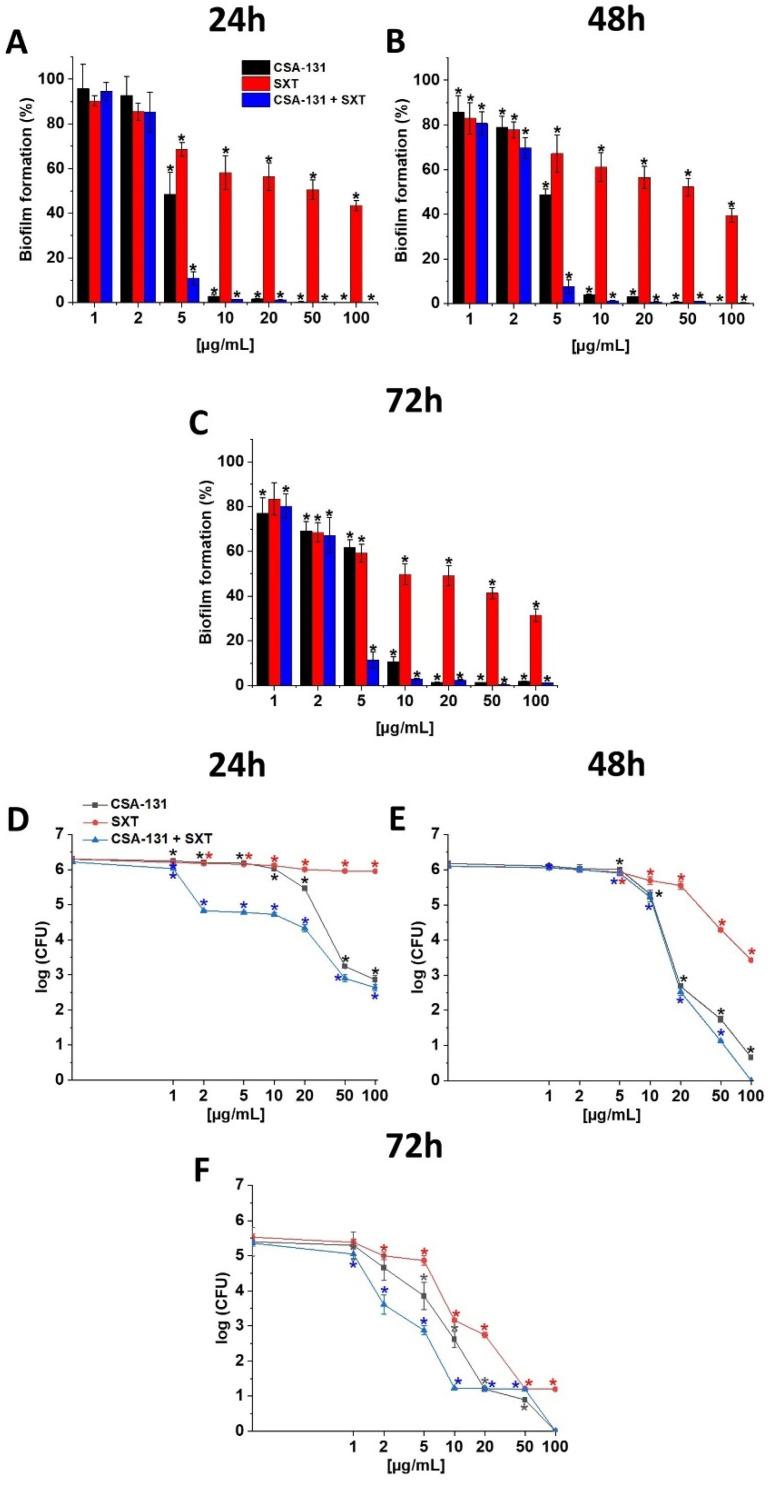
Ability of CSA-131, SXT and their combination to prevent biofilm formation expressed as mean values for *S. melophilia* strains 1, 2, 3, 4 and 5 during treatment with CSA-131, SXT, and CSA-131 + SXT (panels (**A**–**C**)). Disruption of the pre-formed biofilms assessed for strains 1, 2, 3, 4 and 5 after addition of CSA-131, SXT, and CSA-131+SXT (panels (**D**–**F**)). Results show mean ± SD from 3–6 measurements. * indicates statistical significance ≤0.05.

**Figure 3 pathogens-11-00621-f003:**
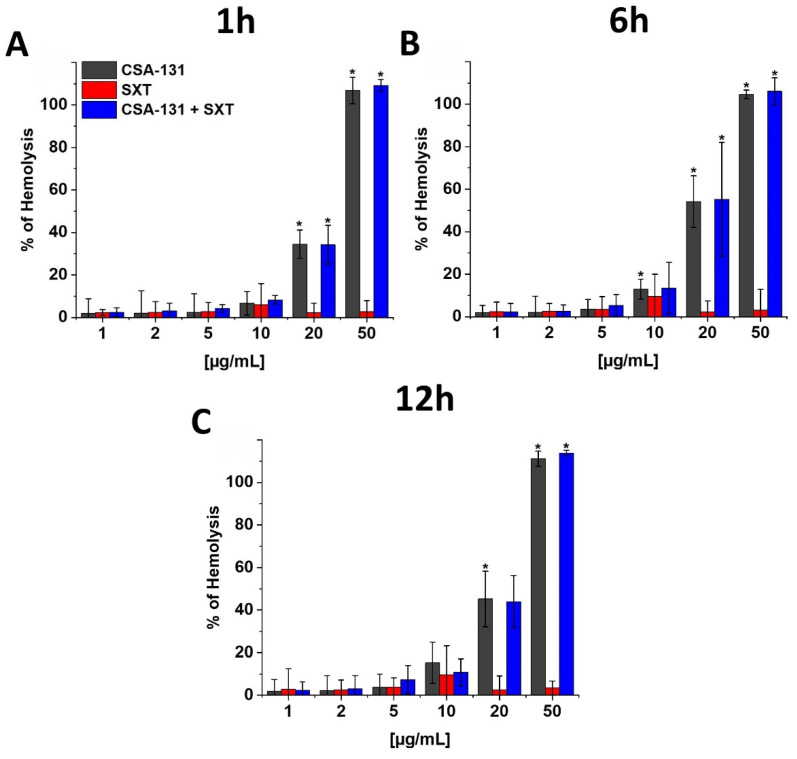
Hemoglobin release from human red blood cells (RBCs) after 1 (panel (**A**)), 6 (panel (**B**)), and 12 (panel (**C**)) hours incubation in the presence of CSA-131, SXT and CSA-131 + SXT ranging from 1–50 μg/mL. Results show mean ± SD, *n* = 3; * indicates statistical significance compared to total hemolysis upon addition of 1% Triton X 100 ≤ 0.05.

**Table 1 pathogens-11-00621-t001:** Antibacterial activity of tested antibiotics and ceragenins against *S. maltophilia* strains. Strains 2–7 are the clinical isolates.

STRAIN NUMBER	Antibiotic	MIC (µg/mL)	MBC (µg/mL)	Ceragenins	MIC (µg/mL)	MBC (µg/mL)
**Strain 1** **Sm (ATCC 13636)**	**CAZ ^1^**	16	32	**CSA-13**	1	2
	**LVX ^2^**	0.5	2	**CSA-44**	4	8
	**SXT ^3^**	4	4	**CSA-131**	2	2
	**COL**	16	16			
**Strain 2**	**CAZ**	256	>256	**CSA-13**	2	4
	**LVX**	4	16	**CSA-44**	2	8
	**SXT**	2	8	**CSA-131**	1	1
	**COL**	2	8			
**Strain 3**	**CAZ**	1	4	**CSA-13**	1	2
	**LVX**	0.25	0.5	**CSA-44**	1	4
	**SXT**	0.5	0.5	**CSA-131**	0.5	2
	**COL**	4	16			
**Strain 4**	**CAZ**	64	64	**CSA-13**	0.5	2
	**LVX**	1	1	**CSA-44**	0.5	2
	**SXT**	0.5	2	**CSA-131**	0.5	1
	**COL**	1	4			
**Strain 5**	**CAZ**	16	32	**CSA-13**	1	4
	**LVX**	1	1	**CSA-44**	2	8
	**SXT**	1	1	**CSA-131**	0.5	1
	**COL**	4	16			
**Strain 6**	**CAZ**	4	16	**CSA-13**	2	4
	**LVX**	0.25	0.5	**CSA-44**	2	8
	**SXT**	0.25	0.5	**CSA-131**	1	4
	**COL**	0.5	2			
**Strain 7**	**CAZ**	32	128	**CSA-13**	2	8
	**LVX**	1	4	**CSA-44**	2	8
	**SXT**	0.5	2	**CSA-131**	2	8
	**COL**	2	16			

^1^ CAZ—ceftazidime, ^2^ LVX—levofloxacin, ^3^ SXT—trimethoprim/sulfamethoxazole (1:19), COL—colistin.

**Table 2 pathogens-11-00621-t002:** MICs values of CAZ, LVX and SXT in combination with CSA-13, CSA-44 and CSA-131 against *S. maltophilia*.

STRAIN	CAZ/ CSA-13 MIC (µg/mL)	CAZ/ CSA-44 MIC (µg/mL)	CAZ/ CSA-131 MIC (µg/mL)	LVX/ CSA-13 MIC (µg/mL)	LVX/ CSA-44 MIC (µg/mL)	LVX/ CSA-131 MIC (µg/mL)	SXT/ CSA-13 MIC (µg/mL)	SXT/ CSA-44 MIC (µg/mL)	SXT/ CSA-131 MIC (µg/mL)
**1** **Sm (ATCC 13636)**	1	1	1	0.5	0.5	1	2	2	2
**2**	4	2	2	1	1	0.5	1	1	0.5
**3**	0.5	1	1	0.5	0.5	0.5	0.125	0.125	0.25
**4**	1	1	0.5	1	1	0.5	0.5	0.25	0.125
**5**	1	4	1	1	1	0.5	0.5	0.5	0.5
**6**	0.5	1	0.5	0.25	0.25	0.125	0.5	0.5	0.25
**7**	4	4	1	1	1	1	0.5	0.5	0.5

**Table 3 pathogens-11-00621-t003:** FICI index in the combination of CAZ, LVX and SXT with CSA-13, CSA-44 and CSA-131 against clinical strains of *S. maltophilia*.

STRAIN	FICI CAZ/ CSA-13	FICI CAZ/CSA-44	FICI CAZ/CSA-131	FICI LVX/CSA-13	FICI LVX/CSA-44	FICI LVX/CSA-131	FICI SXT/CSA-13	FICI SXT/CSA-44	FICI SXT/CSA-131
**1** **Sm (ATCC 13636)**	1.06	0.31	0.56	1.5	1.13	2.5	2.5	1	1.5
**2**	2.02	1	2	0.75	0.75	0.63	1	1	0.75
**3**	1	2	3	2.5	2.5	3	0.38	0.38	1
**4**	2.02	2.02	1	3	3	1.5	2	1	0.5
**5**	1.06	2.25	2.06	2	1.5	1.5	1	0.75	1.5
**6**	0.38	0.75	0.63	1.13	1.13	0.63	2.25	2.25	1.25
**7**	2.13	2.13	0.53	1.5	1.5	1.5	1.25	1.25	1.25

FICI  (<0.5 synergy (S),
<0.76–0.5 is partial synergy (PS), 0.76 < FICI ≤ 1 addition (AD), 1.01 < FICI ≤ 4 is indifference (I), FICI > 4 is antagonism (AN)).

## Data Availability

The data that support the findings of this study are available from the corresponding author upon reasonable request.

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
