# Peer review of "Bactericidal Activity of Ceragenin in Combination with Ceftazidime, Levofloxacin, Co-Trimoxazole, and Colistin against the Opportunistic Pathogen *Stenotrophomonas maltophilia"

_pathogens, 2022, doi:10.3390/pathogens11060621_

Round 1
Reviewer 1 Report
In this study, the authors assessed the susceptibility of selected clinical strains of S. maltophilia to ceragenins: CSA-13, CSA-44 and CSA-131, conventional antibiotics SXT, ceftazidime (CAZ), levofloxacin (LVX) and colistin (COL) and their combinations. The paper is interesting. There are some recommendations to improve the manuscript
- I think the authors did not provide the last version of their manuscript as there are a lot of track changes shown in this version.
- Why do you start with such a high amount of antibiotics ( 256 µg / ml).
- Line 335: 10 µL of the suspension, at which concentration and why you MBC using 10ul.
- Assessment of the fractional index of inhibitory concentration FICI/FIC. Is this the checkerboard assay?
- Combine sections 4.5 and 4.6 together.
- Write MIC and MBC in abbreviation after the first time you put them as full names.
- Why do you use colistin, it is recommended to use colistin due to resistance.
- It is better to show the MIC, and MBC of the individual drug besides the combination for easier comparison.
- Please provide more information bout your future plan.
Author Response
RESPONSES TO REVIEWERS (about manuscript 1734661)
Reviewer 1
Thank you for all your suggestions to improve our manuscript.
1. I think the authors did not provide the last version of their manuscript as there are a lot of track changes shown in this version.
The revision track represents the responses to all 3 Reviewers who rated the original manuscript after first submission
2. Why do you start with such a high amount of antibiotics ( 256 µg / ml).
The experiment was performed accordingly to the recommendations of EUCAST.
3. Line 335: 10 µL of the suspension, at which concentration and why you MBC using 10ul.
These are the recommendations of EUCAST.
4. Assessment of the fractional index of inhibitory concentration FICI/FIC. Is this the checkerboard assay?
FICI/FIC is a checkerboard test.
5. Combine sections 4.5 and 4.6 together.
As requested, the text has been concatenated
6. Write MIC and MBC in abbreviation after the first time you put them as full names.
The abbreviations have been added accordingly.
7. Why do you use colistin, it is recommended to use colistin due to resistance.
Colistin is a last-resort drug and is also a membrane-active antibiotic like CSA.
8. It is better to show the MIC, and MBC of the individual drug besides the combination for easier comparison.
We are sorry, but we are not sure if we understand the Reviewer's point of view well.
9. Please provide more information about your future plan.
We are very excited about our results, so we plan to extend our research to an animal model in the future.
Reviewer 2 Report
The authors have revised the manuscript appropriately.
No further comments.
Author Response
RESPONSES TO REVIEWERS (about manuscript 1734661)
Reviewer 2
The authors have revised the manuscript appropriately.
No further comments.
Thank you very much for your positive comments.
Round 2
Reviewer 1 Report
- Regarding comments 2 and 3, please provide a reference to cite the EUCAST.
- In the tables, it is better to show the MIC, and MBC of the individual drug besides the drug combination for showing the comparison.
- Please elaborate and provide more details about your future studies in the conclusion section.
Author Response
RESPONSES TO EDITOR COMMENTS:
The submitted manuscript is not well prepared. The revision tracks are left in the draft.
Suggest submitting it after revision.
We have accepted all previous changes in the current version of our manuscript
RESPONSES TO REVIEWERS
REVIEWER 1:
- Regarding comments 2 and 3, please provide a reference to cite the EUCAST.
As requested, two ref. were included in revised version of the manuscript
56.EUCAST. Antimicrobial susceptibility testing (AST) of bacteria. MIC determination of non-fastidious and fastidious organisms. Available online: (accessed on
58.EUCAST Definitive Document E.Def 1.2, May 2000: Terminology relating to methods for the determination of susceptibility of bacteria to antimicrobial agents. Clin Microbiol Infect 2000, 6, 503-508, doi:10.1046/j.1469-0691.2000.00149.x.3
Sorry, the MBC was made accordingly to to the CLSI standard
- Clinical and Laboratory Institute, “Methods for Determining Bactericidal Activity of Antimicrobial Agents,” Approved Guideline, CLSI Document M26-A, Wayne, 1999.
Additionally, the same the volume on bacterial solution that was transform on agar plates for determination of MBC was used in some previous study:
- Petrus, E.M, Tinakumari, S, Chai, L. C, Ubong, A, Tunung, R, Elexson, N, Chai, L. F, Son, R. A study on the minimum inhibitory concentration and minimum bactericidal concentration of Nano Colloidal Silver on food-borne pathogens. International Food Research Journal 18: 55-66 (2011)
- Fasolato L, Cardazzo B, Balzan S, et al. Minimum Bactericidal Concentration of Phenols Extracted from Oil Vegetation Water on Spoilers, Starters and Food-Borne Bacteria. Ital J Food Saf. 2015;4(2):4519. Published 2015 May 28. doi:10.4081/ijfs.2015.4519
- C. Hernandes, J. Coppede, B. Bertoni, S. França i A. Pereira, American Journal of Plant Sciences , tom. 4 nr 4, 2013, s. 850-852. doi: 10.4236/ajps.2013.44104 .
- In the tables, it is better to show the MIC, and MBC of the individual drug besides the drug combination for showing the comparison.
Presenting the results of the MIC, MBC and synergistic effect in one table could facilitate interpretation, but in our opinion the table would contain too much data and would not be vague. In our opinion, the current presentation of the results is optimal.
- Please elaborate and provide more details about your future studies in the conclusion section.
We would like to extend our research in an animal model and test the investigated ceragenins, but in combination with nanosystems. The compounds prepared in this way would be highly effective against infections caused by Stenotrophomonas maltophilia. The required changes were included in current version of the manuscript.
This manuscript is a resubmission of an earlier submission. The following is a list of the peer review reports and author responses from that submission.
Round 1
Reviewer 1 Report
General Comments
- Presentation of the results of combination exposure to inhibit or kill undescribed strains of Stenotrophomonas maltophilia.
- The Tables and Figures are confusing and inappropriate and the data must be presented in clearer fashion.
Specific Comments
Abstract
Line 24. Please provide a definition and description of caragenins.
Introduction
Line 39. "actual" is incorrect, please consider current accepted
Lines 62-63. Can be deleted as it is irrelevant
Line 67. Please insert Other, before "Major"
Lines 74-81. Delete as that is not the subject of the manuscript. Also, note that there is no description of the strains in the Materials and Methods.
Lines 82-95. This belongs in the Discussion as it relates to the mechanism of action based on the results derived from the study.
Lines 128-130. This appears to be misleading, as it suggests that no resistance will ever emerge. It would be correct the even long exposure has not resulted in emergence of resistant isolates. Take a look at the citation to be sure the cells were growing.
Lines 131-133. Extraneous and can be deleted.
Results
Lines 143-158 and Table 1. Confusing and completely uninterpretable. In particular Table 1 is impossible to interpret. What is represented by the Column "Ceragenins"? The "Strain Number", namely 1-7 is incomplete, with the exception of the ATCC strain. Please consider the following. For Table 1, a list of antibiotics and ceragenins with MICs against every strain and the hemolytic activity of the same. That would assist with interpretation of MIC/MBCs and possible toxicity.
Lines 160-189 and Table 2. Much of the data simply duplicates what is given in Table 1. The second Table should have only combination data MIC and MBC measurements.
Lines 160-189 and Table 3. I think describing the MIC values (an assumption as the initial data source, MIC or MBC is not identified) as synergistic or inhibitory is overly generous. A factor of 2, considering the variation in MIC values, is simply borderline.
Lines 194-213 and Figure 3. It is inappropriate to present the individual measurements of MBCs (Figure 3) as a line graph. Try a bar chart. Also, there is no note made of the raw data source of the measurements. Either add the table in the text or as a supplement.
Lines 194-213 and Figures 2 DEF. The technique is flawed as it measures both adherence of cells to the wells and then the ensuing growth. Both should be independently measured. As the action of the ceragenins involves a detergent-like mechanism (permeable cells), a likely explanation for the reduction shown in Figure 2 ABC is the reduction of adherence, nothing more.
Lines 214-225 and Figure 3. Please relate the "partial hemolysis" (another undefined term, to the concentrations employed in combination with the antibiotics.
Discussion
Lines 246-269. Background that might be added to the Introduction. Otherwise delete.
Lines 271-284. Inadequate, based on the data. Please focus on actions (i.e., detergent-like activity) that might explain the results.
Materials and Methods
Lines 306-386. No description, save the ATCC strain, of the strains.
Lines 321-334. First, a reference to a standard method is needed and a definition of MIC (partial or complete inhibition) and MBC (90 %, 99 %, 99.5 % etc. reduction in initial colony count).
Lines 335-344. Please provide a definition, in percent or multiple, of FIC and a reference to a well established standard. The reference cited is not a standard.
Lines 345-352. Please provide a definition, in percent or multiple, of FIC and a reference to a well established standard. The reference cited is not a standard.
Lines 353-361. Note that the assay would describe the combined effects of actions on adherence and on growth of attached cells. Biofilm-formation is undefined.
Reviewer 2 Report
In this work, the authors have elucidated the antibacterial role of ceragenins in Stenotrophomonas maltophilia, an opportunistic pathogen that is the causative agent of nosocomial infection in immunocompromised individuals. The authors have tested the different variants of ceragenins CSA-131 showed to be the most potent and it showed to be potent in inhibition of biofilm formation. The study is well conducted, and the writing is clear. The materials and methods have been explained clearly and should be of interest to broader audience studying bacterial pathogenesis and biofilm formation.
I’ve only two minor suggestions:
- The tables 1 and 2 can be a part of the supplementary tables and not a part of main tables.
- The authors showed mentioned if any information is available about the mode of action of the ceragenins. Is there any information about the relative expression of the biofilm formation genes when treated with the ceragenins.
Reviewer 3 Report
In this study, the authors assessed the antimicrobial susceptibility of S. maltophilia. The study is interesting and the manuscript is well written. However, there are few comments that need to be addressed
- The introduction is very long. You need to shorten it a lot and remove nonimportant information.
- You need to make the resistance assay to confirm that this combination will not show resistance.
- Also, it will be better if you do the checkerboard assay and get more insights into the synergism of these compounds.